# NOT-SO-OPTIMAL TRANSPORT FLOWS FOR 3D POINT CLOUD GENERATION

**Ka-Hei Hui**[1]    **Chao Liu**[2]    **Xiaohui Zeng**[2]    **Chi-Wing Fu**[1]    **Arash Vahdat**[2]

[1]The Chinese University of Hong Kong    [2]NVIDIA

https://research.nvidia.com/labs/genair/not-so-ot-flow

## ABSTRACT

Learning generative models of 3D point clouds is one of the fundamental problems in 3D generative learning. One of the key properties of point clouds is their permutation invariance, i.e., changing the order of points in a point cloud does not change the shape they represent. In this paper, we analyze the recently proposed equivariant OT flows that learn permutation invariant generative models for point-based molecular data and we show that these models scale poorly on large point clouds. Also, we observe learning (equivariant) OT flows is generally challenging since straightening flow trajectories makes the learned flow model complex at the beginning of the trajectory. To remedy these, we propose *not-so-optimal transport flow models* that obtain an approximate OT by an offline OT precomputation, enabling an efficient construction of OT pairs for training. During training, we can additionally construct a hybrid coupling by combining our approximate OT and independent coupling to make the target flow models easier to learn. In an extensive empirical study, we show that our proposed model outperforms prior diffusion- and flow-based approaches on a wide range of unconditional generation and shape completion on the ShapeNet benchmark.

## 1 INTRODUCTION

Generating 3D point clouds is one of the fundamental problems in 3D modeling with applications in shape generation, 3D reconstruction, 3D design, and perception for robotics and autonomous systems. Recently, diffusion models (Sohl-Dickstein et al., 2015; Ho et al., 2020) and flow matching (Lipman et al., 2022) have become the de facto frameworks for learning generative models for 3D point clouds. These frameworks often overlook 3D point cloud permutation invariance, implying the rearrangement of points does not change the shape that they represent.

In closely related areas, equivariant optimal transport (OT) flows (Klein et al., 2024; Song et al., 2024) have been recently developed for 3D molecules that can be considered as sets of 3D atom coordinates. These frameworks learn permutation invariant generative models, *i.e.*, all permutations of the set have the same likelihood under the learned generative distribution. They are trained using optimal transport between data and noise samples, yielding several key advantages including low sampling trajectory curvatures, low-variance training objectives, and fast sample generation (Pooladian et al., 2023). Albeit these theoretical advantages, our examination of these techniques for 3D point cloud generation reveals that they scale poorly for point cloud generation. This is mainly due to the fact that point clouds in practice consist of thousands of points whereas molecules are assumed to have tens of atoms in previous studies. Solving the sample-level OT mapping between a batch of training point clouds and noise samples is computationally expensive. Conversely, ignoring permutation invariance when solving batch-level OT (Pooladian et al., 2023; Tong et al., 2023) fails to produce high-quality OT due to the excessive possible permutations of point clouds.

In this paper, we propose a simple and scalable generative model for 3D point cloud generation using flow matching, coined as *not-so-optimal transport flow matching*, as shown in Fig 1. We first propose an efficient way to obtain an approximate OT between point cloud and noise samples. Instead of searching for an optimal permutation between point cloud and noise samples online during training, which is computationally expensive, we show that we can precompute an OT between a dense point superset and a dense noise superset offline. Since subsampling a superset preserves the

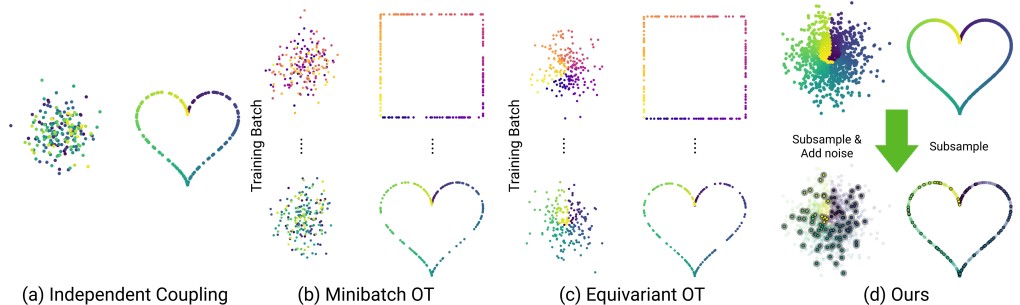

Figure 1: Different coupling types between Gaussian noise (left) and point clouds (right), where coupled noise and surface points share the same color: (a) Independent Coupling randomly maps noises to point clouds. (b) Minibatch OT computes OT map in batches of noises and point clouds. (c) Equivariant OT follows the similar minibatch OT but aligns points via permutation. (d) Our approach precomputes dense OT on data and noise supersets, then subsamples it to couple point clouds with slightly perturbed noise. Note that only (c) and (d) can produce high-quality OT.

underlying shape, we can simply subsample the point superset and obtain corresponding noise from the precomputed OT to construct a batch of noise-data pairs for training the flow models.

We demonstrate that our approximate OT reduces the pairwise distance between data and noise significantly and benefits from the advantages of OT flows, *e.g.*, straightness of trajectories and fast sampling. However, a careful examination shows that learning (equivariant) OT flows is generally challenging since straightening flow trajectories makes the learned flows complex at the beginning of the trajectory. Intuitively, in the OT coupling, the flow model should be able to switch between different target point clouds (i.e., different modes in the data distribution) with small variations in their input, making the flow model have high Lipchitz (see Fig 2).

To remedy this, we propose a simple approach to construct a less "optimal" hybrid coupling by blending our approximate OT and independent coupling used in the flow matching model. In particular, we suggest perturbing the noise samples

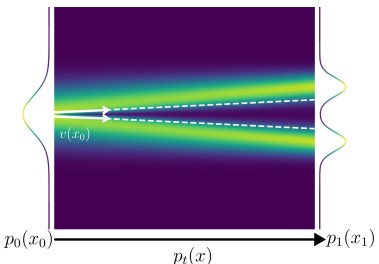

Figure 2: In the OT flow model, the vector field $\mathbf{v}_t(\mathbf{x}_0)$ admits a large change in its output with a small perturbation of $\mathbf{x}_0$ at $t{=}0$.

obtained from our approximate OT with small Gaussian noise. While this remedy makes our mapping less optimal from the OT perspective, we show that it empirically shows two main advantages: First, the target flow model is less complex and the generated points clouds have high sample quality. Second, when reducing the number of inference steps, the generation quality still degrades slower than other competing techniques, indicating smoother trajectories.

In summary, this paper makes the following contributions: (i) We show that existing OT approximations either scale poorly or produce low-quality OT for real-world point cloud generation. (ii) We show that albeit the nice theoretical advantages, equivariant OT flows have to learn a complex function with high Lipchitz at the beginning of the generation process. (iii) To tackle these issues, we propose a not-so-optimal transport flow matching approach that involves an offline superset OT precomputation and online random subsampling to obtain an approximate OT, and adds a small perturbation to the obtained noise during training. (iv) We empirically compare our method against diffusion models, flows, and OT flows on unconditional point cloud generation and shape completion on the ShapeNet benchmark. We show that our proposed model outperforms these frameworks for different sampling budgets over various competing baselines on the unconditional generative task. In addition, we show that we can obtain reasonable generation quality on the shape completion task in less than five steps, which is challenging for other approaches.

## 2 RELATED WORKS

**Score-based Generative Models.** Recently, diffusion models (Sohl-Dickstein et al., 2015; Ho et al., 2020; Song & Ermon, 2019) have gained popularity in generating data of various formats, especially images (Rombach et al., 2022; Ramesh et al., 2022) and videos (Blattmann et al., 2023; Brooks et al., 2024). These generative models employ an iterative process to transform a known distribution,

typically a Gaussian, into a desired data distribution. Song et al. (2020) demonstrate that these models can be generalized to a continuous time setting, solving an SDE with an ODE that shares the same marginal. Lipman et al. (2022) train a vector field to trace a linear interpolation between training pairs from data and noise distribution (further details in Section 3.1). We also focus on the flow matching models and aim to employ them effectively for 3D point cloud generation.

**Relation of Flow Matching with Optimal Transport (OT).** Flow matching is closely related to the Optimal Transport (OT) problem, which aims to find a map with minimal transport cost between two distributions. As for flow matching, the trajectory flows can be taken to define the map that solves the OT problem. However, when the flow models are trained with random pairs, the trajectories are usually curved, so using pairs from the OT map can lead to straighter trajectories that improve the efficiency, as Pooladian et al. (2023); Tong et al. (2023) show. Since the OT map is usually unavailable, various methods are introduced to obtain better training pairs as we discuss in Section 3.1. Another approach to obtain better pairs is to iteratively straighten the trajectory from a flow matching model. Rectified Flow (Liu et al., 2022) obtains training pairs by simulating ODE trajectories of pre-trained models. However, this involves an expensive ODE simulation and introduces errors in the straightening process. In this work, we mainly study the limitations of existing OT solutions on the first approach and derive a new solution that is specific for point cloud generation.

**3D Point Cloud Generation.** Point cloud generation has been studied extensively using various generative models including VAEs (Kim et al., 2021), Energy-based models (Xie et al., 2021), GANs (Achlioptas et al., 2018; Shu et al., 2019; Li et al., 2021), and flow-based models (Yang et al., 2019; Kim et al., 2020). Recently, diffusion models (Zeng et al., 2022; Zhou et al., 2021; Luo & Hu, 2021; Peng et al., 2024; Zhao et al., 2025) have achieved great success in producing high-quality generation results. However, these methods often ignore point clouds' permutation-invariant nature in training by treating them as high-dimensional flat vectors. PSF (Wu et al., 2023) uses Rectified flow (Liu et al., 2022) to generate point clouds, addressing permutation issues implicitly through flow straightening. This approach is computationally expensive due to multiple sample inferences needed for constructing training pairs. In this work, we focus on constructing an efficient, high-quality OT approximation for permutation-invariant point cloud generation based on the flow matching models.

# 3 METHOD

In Section 3.1, we begin by covering preliminaries of training a continuous normalizing flow and recent OT flows. Section 3.2 explores the challenges of applying existing OT approximation methods to 3D point clouds. To tackle these challenges, we introduce our approximate OT approach in Section 3.3 that precomputes OT maps in an offline fashion, and in Section 3.4, we explore a simple hybrid and less optimal coupling approach that makes target flows easier to learn.

## 3.1 PRELIMINARIES

**Continuous Normalizing Flow (CNF)** (Chen et al., 2018) morph a base Gaussian distribution $q_0$ into a data distribution $q_1$ using a time-variant vector field $\mathbf{v}_{\theta,t} : [0, 1] \times \mathbb{R}^d \to \mathbb{R}^d$, parameterized by a neural network $\theta$. The mapping is obtained from an ordinary differential equation (ODE):

$$\frac{d}{dt}\mathbf{x}_t = \mathbf{v}_{\theta,t}(\mathbf{x}_t). \tag{1}$$

Conceptually, the ODE transports an initial sample $\mathbf{x}_0 \sim q_0$, where $\mathbf{x}_0 \in \mathbb{R}^d$ with $p_t$ denoting the distribution of samples at step $t$ and $p_0(\mathbf{x}) := q_0(\mathbf{x})$. Usually, the vector field $\mathbf{v}_{\theta,t}$ is trained to maximize the likelihood $p_1$ assigned to training data samples $\mathbf{x}_1$ from distribution $q_1$. This procedure is computationally expensive due to extensive ODE simulation for each parameter update.

**Flow Matching** (Lipman et al., 2022) avoid the computationally expensive simulation process for training CNFs. In particular, we define a conditional vector field $\mathbf{u}_t(\cdot|\mathbf{x}_1)$ and path $p_t(\cdot|\mathbf{x}_1)$ that transform $q_0$ into a Dirac delta at $\mathbf{x}_1$ at $t = 1$. Lipman et al. (2022) show that $\mathbf{v}_{\theta,t}$ can be learned via a simple conditional flow matching (CFM) objective:

$$\mathcal{L}_{\text{CFM}} = \mathbb{E}_{t,q_1(\mathbf{x}_1),q_0(\mathbf{x}_0)}||\mathbf{v}_{\theta,t}(\mathbf{x}_t) - \mathbf{u}_t(\mathbf{x}_t|\mathbf{x}_1)||^2. \tag{2}$$

A common choice for the conditional vector field is $\mathbf{u}_t(\mathbf{x}|\mathbf{x}_1) := \mathbf{x}_1 - \mathbf{x}_0$, which can be easily simulated by linearly interpolating the data and Gaussian samples via $\mathbf{x}_t = (1 - t) * \mathbf{x}_0 + t\mathbf{x}_1$.

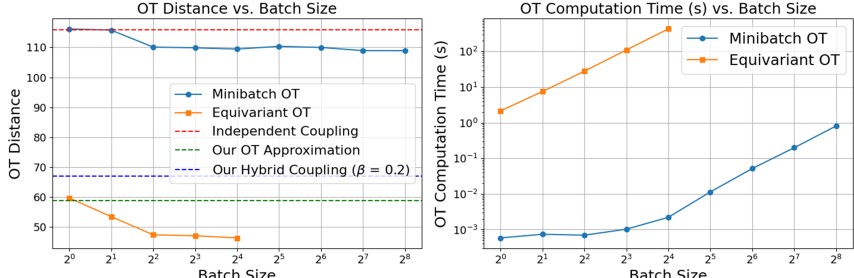

Figure 3: Comparison of OT Approximation Methods. **Left:** Average OT distance across batch sizes. Minibatch OT (blue) fails to reduce distances much compared to independent coupling (red dash). Equivariant OT (orange) significantly reduces distance values. Our OT approximation is on par with Equivariant OT. **Right:** Computational time for OT across batch sizes. Minibatch OT (blue) maintains a reasonable computational time ($\sim$1 second) with batch size $B = 256$. Equivariant OT (orange) grows quadratically starting from 2.2 seconds with $B = 1$.

**Optimal Transport (OT) Map.** In the CFM objective in Eq. 2, the training pair $(\mathbf{x}_0, \mathbf{x}_1)$ is sampled from an independent coupling: $q(\mathbf{x}_0, \mathbf{x}_1) = q_0(\mathbf{x}_0)q_1(\mathbf{x}_1)$. However, Tong et al. (2023); Pooladian et al. (2023) show that we can sample the training pair from any coupling that satisfies the marginal constraint: $\int q(\mathbf{x}_0, \mathbf{x}_1)d\mathbf{x}_1 = q_0(\mathbf{x}_0)$ and $\int q(\mathbf{x}_0, \mathbf{x}_1)d\mathbf{x}_0 = q_1(\mathbf{x}_1)$. They show an optimal transport (OT) map $\pi$ that minimizes $\int ||\mathbf{x}_0 - \mathbf{x}_1||^2 \pi(\mathbf{x}_0, \mathbf{x}_1)d\mathbf{x}_0 d\mathbf{x}_1$ is a good choice for data coupling, leading to a straighter trajectory. Yet, obtaining the optimal transport map is often difficult for complex distributions. Next, we review two main directions for approximating the OT map:

**(i) Minibatch OT:** Tong et al. (2023); Pooladian et al. (2023) approximate the actual OT by computing it at the batch level. Specifically, they sample a batch of Gaussian noises $\{\mathbf{x}_0^1, \cdots, \mathbf{x}_0^B\} \sim q_0$ and data samples $\{\mathbf{x}_1^1, \cdots, \mathbf{x}_1^B\} \sim q_1$, where $B$ is the batch size. They solve a discrete optimal transport problem, assigning noises to data samples while minimizing a cost function $C(\mathbf{x}_0, \mathbf{x}_1)$. The cost function is typically the squared-Euclidean distance, *i.e.*, $C(\mathbf{x}_0, \mathbf{x}_1) = ||\mathbf{x}_0 - \mathbf{x}_1||^2$, and the assignment problem is often solved using the Hungarian algorithm (Kuhn, 1955). After computing the assignment, we can use the assigned pairs to train the vector field network via Eq. 2. As the batch size $B$ approaches infinity, this procedure converges to sampling from the true OT map.

**(ii) Equivariant OT Flow Matching:** Song et al. (2024); Klein et al. (2024) also approximate the OT at the batch level, but they focus on generating elements invariant to certain group $G$, such as permutations, rotations, and translations. Specifically, they propose replacing the aforementioned cost function $C(\mathbf{x}_0, \mathbf{x}_1)$ with one that accounts for these group elements: $C(\mathbf{x}_0, \mathbf{x}_1) = \min_{g \in G} ||\mathbf{x}_0 - \boldsymbol{\rho}(g)\mathbf{x}_1||^2$, where $\boldsymbol{\rho}(g)$ is the matrix representation of the group element $g$. This approach significantly reduces the OT distance even with a small batch size, demonstrating success in generating molecular data. Intuitively, using the cost function defined above allows us to align data and noise together (in our case via permutation) when computing the minibatch OT.

So far, we consider generic unconditional generative learning. It is worth noting that mini-batch OT does not easily extend to conditional generation problems, *i.e.*, learning $p(\mathbf{x}|\mathbf{y})$ for a generic input conditioning $\mathbf{y}$, when there is only one training sample $\mathbf{x}$ for each input conditioning $\mathbf{y}$. This is because the OT assumes access to a batch of training samples for each $\mathbf{y}$.

### 3.2 EXISTING OT APPROXIMATION FOR POINT CLOUD GENERATION

We focus on generating 3D shapes represented as point clouds. A point cloud $\mathbf{x}_1 \in \mathcal{R}^{N \times 3}$ is a set of points sampled from the surface of a shape $\mathcal{S}$, where $N$ is the number of points. Unlike 2D images, point clouds have unique properties that pose challenges for existing OT methods:

**(i) Permutation Invariance.** A point cloud, while arranged in a matrix form, is inherently a set. Shuffling points in $\mathbf{x}_1$ should represent the same shape. Mathematically, given a permutation matrix $\boldsymbol{\rho}(g)$, the sampling probability remains unchanged, *i.e.*, $q_1(\boldsymbol{\rho}(g)\mathbf{x}_1) = q_1(\mathbf{x}_1)$.

**(ii) Dense Point Set.** Point clouds are finite samples on surfaces. However, similar to low-resolution images, sparse point sets may miss fine geometric structures and details. Thus, most works use dense point sets (say $N \geq 2048$) to accurately capture 3D shapes.

Existing approach to estimating OT maps face these challenges on point clouds:

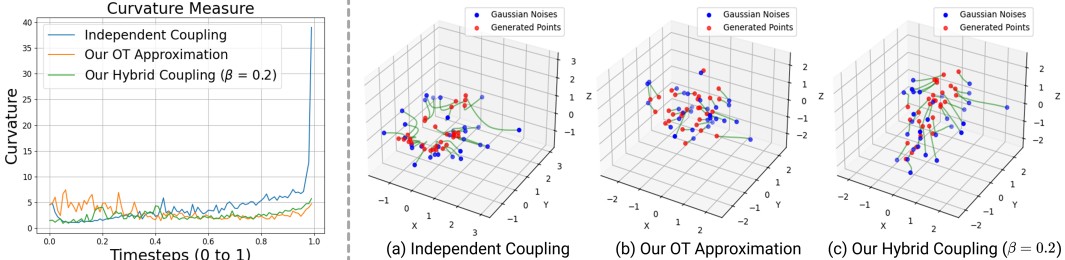

Figure 4: Analysis of trajectory straightness using different couplings to obtain training pairs. Left: We plot the square norm of the difference between successive vector fields, i.e., $\|v_{\theta,t+1}(x_{t+1}) - v_{\theta,t}(x_t)\|$, as a measure of trajectory curvature. Right (a-c): Trajectory samples obtained by models trained with (a) independent coupling, (b) our OT approximation, and (c) our hybrid coupling with $\beta = 0.2$. Note that we subsample the point cloud to 30 points for a better trajectory visualization.

**Ineffectiveness of Minibatch OT.** Minibatch OT, effective in low-dimensional and image domains, fails for point clouds due to property (i). There are $N!$ equivalent representations of the same point cloud, implying $N!$ equivalent training pairs $(\mathbf{x}_0, \mathbf{x}_1)$. An OT-sampled pair should minimize the cost: $C(\mathbf{x}_0, \mathbf{x}_1) = \min_{g \in G} C(\mathbf{x}_0, \boldsymbol{\rho}(g)\mathbf{x}_1)$. However, in Minibatch OT's with no permutation, the assignments grow quadratically with batch size, while the number of permutations grows exponentially. As shown in Figure 3 (left), Minibatch OT achieves only about 6% reduction in the cost even with batch size 256, indicating limited effectiveness of this approach in point cloud generation.

**Inefficient OT Maps in Equivariant OT.** Equivariant OT produces high-quality maps, but is computationally expensive for point cloud generation. Figure 3 (left) shows a 48.7% reduction even with a batch size 1, showing the importance of aligning points and noise via permutation. However, unlike molecular data, which has limited size ($N$=55 in (Klein et al., 2024)), representing 3D shapes needs a larger $N$, following property (ii). Solving the optimal transport cost takes an $O(B^2 N^3)$ computational complexity because of the quadratic number of noise and point cloud pairs in a batch of $B$ examples, and $O(N^3)$ for the the Hungarian algorithm (Kuhn, 1955). Figure 3 (right) shows how this grows rapidly even for a typical point cloud size ($N = 2048$). It takes around 2.2 seconds for the OT computation even for $B = 1$, leading to a significant bottleneck in the training process that is more than 40x slower than independent coupling.

### 3.3 OUR OT APPROXIMATION

A simple approach to generate training point clouds is to re-sample the points from the object surface in each training iteration. However, most point cloud generation methods avoid this tedious online sampling by pre-sampling a dense point superset $X_1 \in \mathcal{R}^{M \times 3}$ with $M >> N$. During training, random subsets of $X_1$ are selected as training targets. This procedure converges to the true sampling distribution, following a straightforward extension of the law of large numbers (see Appendix proposition 1 for details). In a similar spirit, we compute an offline OT map between a dense point superset $X_1 \in \mathcal{R}^{M \times 3}$ and a dense randomly-sampled Gaussian noise superset $X_0 \in \mathcal{R}^{M \times 3}$, and during training, subsample data-noise pairs from the supersets based on the offline OT map.

**Superset OT Precomputation.** Given supersets $X_0$ and $X_1$, we compute a bijective map $\Pi$ between them, i.e., $\Pi : X_0 \leftrightarrow X_1$. When $M$ is small, following (Song et al., 2024; Klein et al., 2024), we compute the bijective map using the Hungarian algorithm (Kuhn, 1955). However, this algorithm scales poorly for large point clouds, i.e., $M > 10K$. For such large point clouds, we use Wasserstein gradient flow to transform $X_0$ into $X_1$ by minimizing their Wasserstein distance iteratively. Using an efficient GPU implementation (Feydy et al., 2019), the OT precomputation takes only ~30 seconds for 100K points, showing its high scalability. See Appendix C for the details in procedure.

**Online Random Subsampling.** Given precomputed coupling $\Pi(X_0, X_1)$, we randomly sample data-noise pair $(\mathbf{x}_0, \mathbf{x}_1) \sim \Pi(X_0, X_1)$ and we train the flow matching model according to Equation 2. As we show in Figure 3 (left), this significantly reduces the transport cost, while introducing negligible training overhead. In Appendix A.1.2, we show that the sampled training pair converges to correct marginals if $M$ is sufficiently large.

In practice, using these pairs for training results in straighter sampling trajectories, measured by the curvature of the sampling trajectory, as shown in Figure 4 (left). The model trained with our OT approximation (orange curve) exhibits a much lower maximum curvature compared to the one with

independent coupling (blue curve). We also visualize the sample trajectories of these two models in Figures 4 (right) (a-b), confirming straighter trajectories for our model.

### 3.4 HYBRID COUPLING

Though training flows with OT couplings comes with appealing theoretical justifications (*e.g.*, straight sampling trajectories), we identify a key training challenge with them that is often overlooked in the OT flows literature. Our experiments (*e.g.*, Section 4.1) indicate that flows trained with equivariant OT maps are often outperformed by those with independent coupling in terms of sample quality, especially when the number of sampling steps is large. We hypothesize this is due to the increasing complexity of target vector fields for OT couplings that makes their approximation harder with neural networks.

Intuitively, as we make target sampling trajectories straighter using more accurate OT couplings, the complexity of generation shifts toward smaller time steps. As Figure 2 shows, in the limit of straight trajectories, the learned vector field $\mathbf{v}_{\theta,0}(\mathbf{x}_0)$ should be able to switch between different target point clouds with small variation in $\mathbf{x}_0$, forcing $\mathbf{v}_{\theta,0}$ to be complex at $t=0$. This problem is further exacerbated in the equivariant OT flows with large $N$ where permuting Gaussian noise cloud in the input makes it virtually the same for all target point clouds. To verify this, we measure the trained vector field's complexity for 3D point cloud generation using the Jacobian Frobenius norm in different timesteps in Figure 5. As hypothesized above, switching from independent coupling (blue curve) to our OT approximation (orange curve) shifts the high Jacobian norm at $t \approx 1$ for independent coupling to $t \approx 0$ for OT coupling. This motivates us to develop a method to make it easier

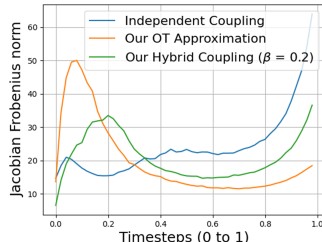

Figure 5: We show Jacobian Frobenius Norm for different trained $\mathbf{v}_{\theta,t}$ over different time intervals, which measures the model complexity as in (Dockhorn et al., 2022).

for neural networks to approximate the target vector field, while still maintaining a relatively straight path.

**Hybrid Coupling.** Given the different behavior of independent and OT couplings in Figure 5, we aim to reduce the complexity of the vector field at early timesteps by combining our OT approximation with independent coupling. To do so, we propose injecting additional random Gaussian noise into $\mathbf{x}_0$, making our OT couplings even less "optimal". The new training $\mathbf{x}_0'$ is defined as:

$$\mathbf{x}_0' = \sqrt{1-\beta}\mathbf{x}_0 + \sqrt{\beta}\boldsymbol{\epsilon}, \quad \boldsymbol{\epsilon} \sim \mathcal{N}(\boldsymbol{\epsilon}; 0, \mathbf{I}), \tag{3}$$

where $\beta \in [0, 1]$ is the blending coefficient. Intuitively $\beta$ allows us to switch smoothly between independent and OT couplings. Specifically, for $\beta \to 0$, the coupled data and noise pairs converge to our OT couplings, whereas when $\beta \to 1$, they follow the independent coupling.

We empirically observe that $\beta = 0.2$ in most experiments strikes a good balance between learning complexity (as shown by the green curve in Figure 5), low curvature for the sampling trajectories (the green curve in Figure 4 (left) and Figure 4 (right) (c)), and sample generation quality (shown later in Section 4.3). In the next section, we refer to this hybrid coupling as our main method.

### 4 EXPERIMENT

In this section, we present our experimental results for unconditional and conditional 3D point cloud generation, *i.e.*, shape completion, after reviewing dataset and evaluation details.

**Dataset.** Following Yang et al. (2019); Klokov et al. (2020); Cai et al. (2020); Zhou et al. (2021), we employ the ShapeNet dataset (Chang et al., 2015) for training and evaluating our approach. Specifically, we train separate generative models for the Chair, Airplane, and Car categories with the provided train-test splits. Additionally, we provide more visualizations in Appendix on training a single model on all three categories to validate the generalization of our approach. To form our training point clouds, we randomly sample a superset of $M = 100K$ points on each input 3D shape. Then, during the online random subsampling, we randomly subsample the superset to $N = 2,048$ points, following the procedure in Section 3.3.

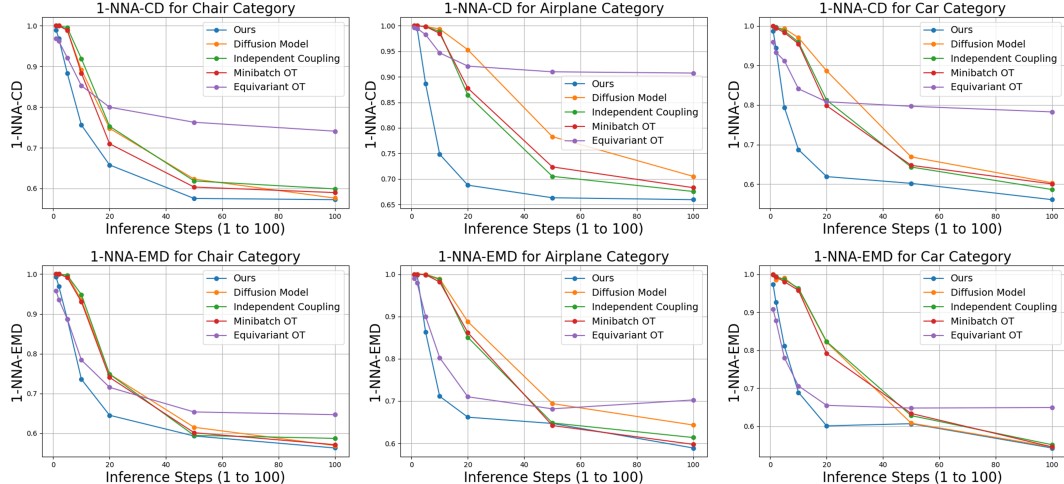

Figure 6: Quantitative comparisons of generation quality for different training paradigms using 1-NNA-CD (top) and 1-NNA-EMD (bottom) for Chair (left), Airplane (middle), and Car (right). We present evaluation metrics across various inference steps, *i.e.*, from 1 steps to 100 steps, for five methods: (i) ours, (ii) diffusion model with v-prediction (Salimans & Ho, 2022), and three flow matching models with different coupling methods: (iii) independent coupling (Lipman et al., 2022), (iv) Minibatch OT (Tong et al., 2023; Pooladian et al., 2023), and (v) Equivariant OT (Song et al., 2024; Klein et al., 2024). Note that values closer to $50\%$ indicate better performance.

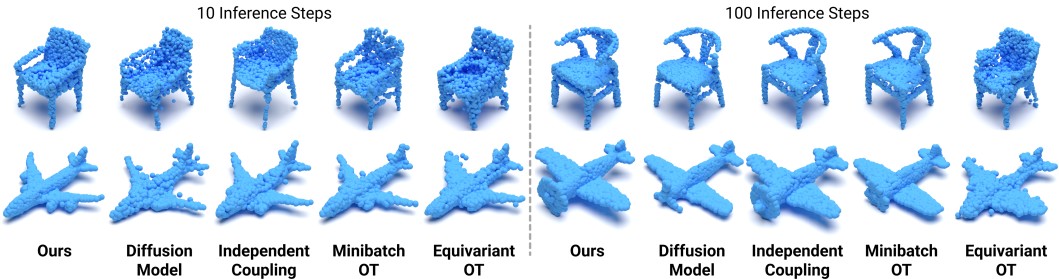

Figure 7: Qualitative comparisons of generation quality for Chair (top) and Airplane (bottom) categories. We present inference results with 10 steps (left) and 100 steps (right).

In addition, we prepare a partial input shape for the shape completion task for each training shape. We use the GenRe dataset (Zhang et al., 2018) for depth renderings of ShapeNet shapes. Partial point clouds are obtained by unprojecting and sampling up to 600 points from these depth images.

**Evaluation Metrics.** We use LION (Zeng et al., 2022)'s evaluation protocol, focusing on 1-NN classifier accuracy (1-NNA) with Chamfer Distance (CD) and Earth Mover's Distance (EMD) metrics. 1-NNA measures how well the nearest neighbor classifier differentiates generated shapes from test data. Optimal generation quality is achieved when the classifier accuracy is $\sim 50\%$. As discussed by Yang et al. (2019), 1-NNA is more robust and correlates better with generation quality.

For shape completion, since we have a ground-truth (GT) shape for each given condition, we follow (Zhou et al., 2021) to measure the similarity of our generated shape against the GT shape. In particular, we randomly sample 2,048 points on the GT shape surface and use CD and EMD distance metrics, where lower values indicate a higher similarity and thus a better performance.

## 4.1 UNCONDITIONAL GENERATION

For unconditional generation, we first evaluate our framework against alternative training paradigms, including diffusion models and flow matching models with different couplings:

**Baselines.** We consider both quantitative and qualitative comparisons. In this setting, We maintain the same architecture and hyperparameters, changing only the training procedure or objective for fair comparison. We then compare with four training alternatives: (i) the diffusion model objective using v-prediction (Salimans & Ho, 2022), and (ii)-(iv) three flow matching models with different coupling methods: independent coupling (Lipman et al., 2022), Minibatch OT (Tong et al., 2023;

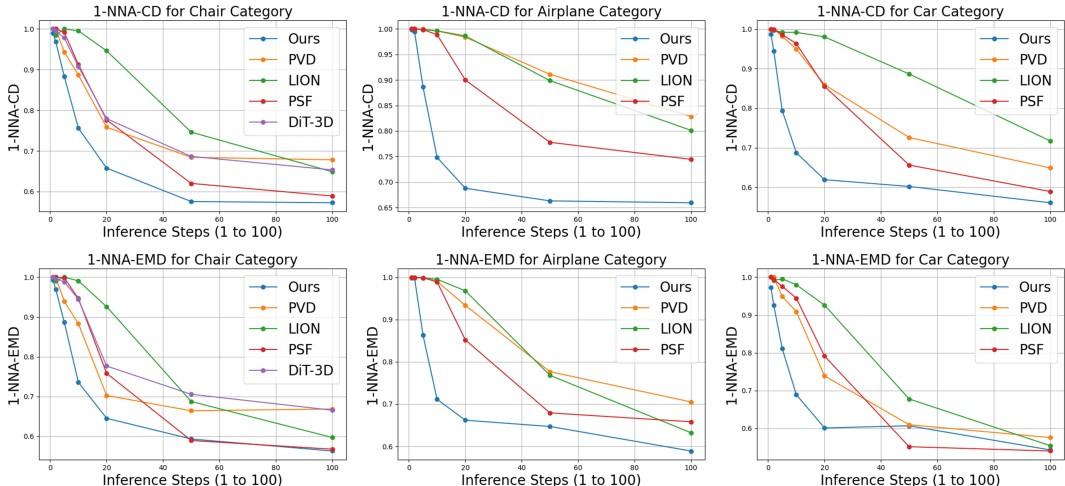

Figure 8: Quantitative comparisons with other point cloud generation methods using 1-NNA-CD (top) and 1-NNA-EMD metrics (bottom) for Chair (left), Airplane (middle), and Car (right). We present evaluation metrics across various inference steps, *i.e.*, from 1 step to 100 steps, for five methods: (i) ours, (ii) PVD (Zhou et al., 2021), (iii) LION (Zeng et al., 2022), (iv) PSF (Wu et al., 2023) without rectified flow, and (v) DiT-3D (Mo et al., 2023).

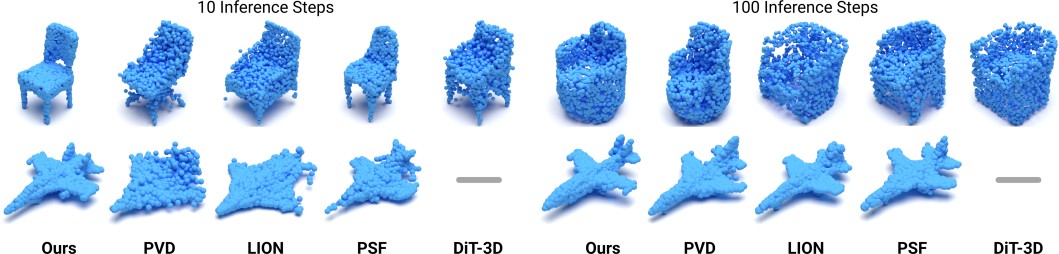

Figure 9: Qualitative comparisons of generation quality for Chair (top) and Airplane (bottom) categories. We present inference results with 10 steps (left) and 100 steps (right).

Pooladian et al., 2023), and (iv) Equivariant OT (Song et al., 2024; Klein et al., 2024). For Minibatch OT, we obtain the OT using a batch size of 64 on each GPU but compute the training losses across all four GPUs. For Equivariant OT, due to its high computational demand, we can only consider permutations with a batch size of 1, and train the model with the same training time (4 days).

**Quantitative Comparisons.** Figure 6 plots 1-NNA-CD & EMD for varying computation budget (inference steps) over the Chair, Airplane, and Car categories. Overall, our approach (blue curve) achieves similar or better performance across all metrics and categories, particularly when given a sufficient number of steps, *e.g.*, 100. Our approach performs best with fewer inference steps (10-20) due to a straighter trajectory, demonstrating the advantages of our approximate OT. Minibatch OT's generation performance matches original flow matching, possibly due to small OT distance reductions between training pairs. Equivariant OT (orange curve) shows inferior performance compared to others, likely due to slow training within training budget.

**Qualitative Comparisons.** Figure 7 shows visual comparisons of different training methods for the Chair and Airplane categories, which feature more distinguishable characteristics. To facilitate easier comparisons, we use our generation results to retrieve the closest results from other methods. We display the generation results with 10 inference steps on the left. Unlike other methods that often add noise to distinct shape parts such as chair's armrests or airplane's wings, our approach better preserves these structures. With sufficient steps (right), nearly all methods (except Equivariant OT) can produce high-quality results with thin structures and fine details, *e.g.*, the back of the chair.

Next, we compare with other point cloud generation methods that require multi-steps generation:

**Baselines.** We also compare our framework with the following four baselines under the same inference budget: (i) PVD (Zhou et al., 2021) and (ii) DiT-3D (Mo et al., 2023) are diffusion-based models that directly generate point clouds, whereas (iii) LION (Zeng et al., 2022) employs a diffusion model to produce a set of latent points that are later decoded into a point cloud. Since these

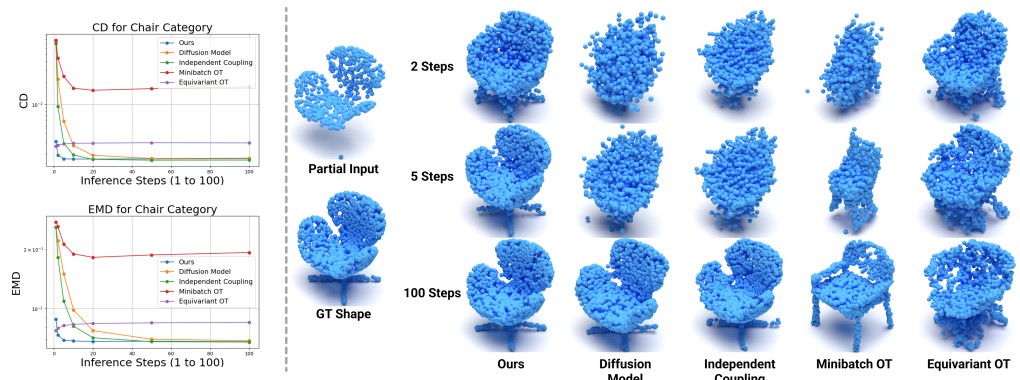

Figure 10: Comparisons with other training paradigms on the shape completion task. Left: Quantitative comparisons with other alternatives using different inference steps. Right: Qualitative comparisons with other methods show the completion generated by 2, 5, and 100 steps, respectively.

models are trained for more inference steps (1,000), we employ a DDIM sampler at inference, which is considered to be effective with fewer inference steps. (iv) Lastly, PSF (Wu et al., 2023), similar to our approach, uses a flow-based generative model, but additionally applies a rectified flow procedure (Liu et al., 2022) to progressively straighten the inference trajectory, taking significantly additional training time. Since our method can be further accelerated with rectified flows, we only compare performance with PSF before the expensive rectification step. Note that we only report DiT-3D's result on the Chair category, as the pre-trained models for other categories are unavailable.

**Quantitative Comparisons.** Figure 8 shows results on 1-NNA-CD & EMD for different inference steps. Our method shows a comparable performance with existing flow-based and diffusion-based generative models when given 100 inference steps. Additionally, our method can achieve significantly better generation quality when the inference is reduced to around 20 steps, without relying on additional expensive straightening procedures, *e.g.*, rectified flow.

**Qualitative Comparisons.** Figure 9 shows visual comparisons with other point cloud generation methods for the Chair and Airplane categories, which feature more distinguishable characteristics. Overall, all existing methods can generate high-quality 3D shapes when given a large number of inference steps. However, when the number of inference steps is reduced, the generation quality of other methods declines significantly, as illustrated by the jet planes example on the left.

## 4.2 SHAPE COMPLETION

**Baselines.** We evaluate against the same baselines from Figure 6 for the shape completion task.

**Quantitative Comparisons.** Figure 10 (left) shows the evaluation metrics (CD & EMD) against various inference steps. We observe that our model can achieve reasonable generation quality with only five inference steps, while other approaches typically require about 50 steps to obtain similar quality. For Equivariant OT, it can produce comparable performance to our approach at two inference steps but fails to improve further with additional steps. Minibatch OT shows poor metrics, even with inference steps, as its batch-level OT is computed under different partial shapes. This violates assumptions that OT should be computed between training instances with the same condition.

**Qualitative Comparisons.** We present a visualization of shape completion results for various methods in Figure 10 (right). Our approach produces a reasonable-looking shape with only five inference steps, while other methods still generate noisy shapes. Furthermore, Minibatch OT generates a shape that does not correspond to the input due to violating the training assumption. Equivariant OT fails to produce shapes of similar visual quality even with increased inference steps.

## 4.3 MODEL ANALYSIS

In the followings, we perform analysis on major modules in our approach, including the blending coefficient $\beta$ of the hybrid coupling and the effect of the superset size $M$ for the OT computation.

**Blending Coefficient** $\beta$**.** We examine the impact of different couplings on flow matching model training. Specifically, we train models with coupling blending coefficient $\beta$ from 0 (OT approximation) to 1.0 (independent coupling). For each value, we train a model from scratch on the Chair Category and evaluate it using the 1-NNA-CD & EMD metrics with 100 steps. To avoid approximation errors in larger supersets, we adopt a superset size of $10,000$ with exact OT in this experiment.

Table 1: 1-NNA-CD & EMD over 100 steps for models trained with different $\beta$.

| Metrics | Interpolation Coefficient ($\beta$) | | | | | | |
|---|---|---|---|---|---|---|---|
| | 0 | 0.001 | 0.01 | 0.1 | 0.2 | 0.5 | 1.0 |
| 1-NNA-CD | 0.7289 | 0.6918 | 0.6223 | 0.5968 | **0.5853** | 0.5861 | 0.5989 |
| 1-NNA-EMD | 0.6647 | 0.6563 | 0.6156 | 0.5884 | **0.5741** | 0.5780 | 0.5869 |

Table 2: 1-NNA-CD & EMD over 100 steps for models trained using different superset sizes ($M$).

| Metrics | OT Superset Size ($M$) | | | | | |
|---|---|---|---|---|---|---|
| | 2048 | 5000 | 10000 | 20000 | 50000 | 100000 |
| 1-NNA-CD | 0.6352 | 0.5853 | 0.5853 | 0.5853 | 0.5921 | **0.5725** |
| 1-NNA-EMD | 0.6254 | 0.5853 | 0.5741 | **0.5627** | 0.5695 | **0.5627** |

Table 1 presents the results. We can observe that directly employing our OT approximation ($\beta = 0.0$) can lead to inferior performance, which can be gradually improved by injecting a small amount of noise into the coupling. Interestingly, the best performance is achieved when compared with other cases at around $\beta = 0.2$. Injecting more noise until arriving at independent coupling does not yield further improvement. These results demonstrate that neither our OT approximation nor independent coupling is optimal in terms of generation quality, and the hybrid coupling is necessary.

**OT Supersets Size $M$.** Next, we investigate the importance of constructing sufficiently large supersets for OT computation. Here, we try supersets of varying sizes, starting from 2,048 (number of points to be generated) and progressively increasing the number towards 100,000. As outlined in Section 3.3, we employ the exact OT method for superset sizes of 10,000 or fewer points, and the OT approximation method for larger sets. For each superset size, we also train a flow matching model on the Chair Category and evaluate also its generation quality over 100 inference steps.

Table 2 reports the evaluation results. Overall, we notice that a small superset size usually leads to slightly worse performance, potentially due to overfitting the same target generation points. Increasing the number of points in the superset helps improve the performance. Notably, despite using an approximate OT (introduced in Section 3.3), we still observe some improvement in the generation quality when using a large superset ($M = 100,000$), indicating the benefit of using a large superset outweighs the errors introduced by the approximation.

## 5 CONCLUSION

In conclusion, we proposed a novel framework, coined as not-so-optimal transport flow matching for 3D point cloud generation. We demonstrated that existing OT approximation methods scale poorly for large point cloud generation and showed that target OT flow models tend to be more complex, and thus, harder to be approximated by neural networks. To address these challenges, our approach introduces an offline superset OT precomputation followed by an efficient online subsampling. To make the target flow models less complex, we proposed hybrid coupling that blends our OT approximation and independent coupling, making our OT intentionally less optimal. We demonstrated that our proposed framework can achieve generation quality on par with existing works given sufficient inference steps, while achieving superior quality with smaller sampling steps. Additionally, we show our approach is effective for conditional generation tasks, such as shape completion, achieving good generation even with five steps. Despite these advancements, there are still several limitations and potential extensions of this work. First, we do not consider other forms of invariance, such as rotational invariance that is present in large shape datasets such as Objaverse (Deitke et al., 2023). Second, we only consider point clouds representing coordinates. It would be interesting to explore the generation of point clouds with additional information, such as points with colors or even Gaussian splitting. Third, we currently focus on generation with a fixed resolution, and it would be interesting to extend our method for resolution-invariant cases, such as those in conditional tasks (Huang et al., 2024). Forth, we assume our dataset does not contain outliers in the point cloud following existing works, and developing a robust learning procedure with a noisy point cloud dataset would be another interesting direction. Fifth, we show empirically that hybrid coupling can reduce the trained models' complexity (as shown in Figure 5), a theoretical connection between $\beta$ and the complexities is yet to be studied. Last but not least, we hope that this work can inspire further development around OT maps that are easier to learn and that our proposal, especially for the hybrid coupling, can translate to success at generating other forms of large point clouds such as large molecules or proteins.

## 6 REPRODUCIBILITY STATEMENT

We demonstrate our efficiency in enabling the reproduction of this work's results. In particular, we describe the dataset (ShapeNet (Chang et al., 2015)) and the data splits employed in training and evaluation of this work in Section 4. We also cover the evaluation protocol used in this work for assessing performance. Additionally, we include implementation details, such as training specifics, network architecture, and shape normalization for evaluation in Appendix B. Lastly, we outline the procedure of our superset computation and describe the hyperparameter choices in Appendix C.

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

# A  APPENDIX

## A.1  PROOF FOR SATISFACTION OF MARGINAL CONSTRAINTS.

### A.1.1  LAW OF LARGE NUMBERS

**Proposition 1.** *Given* $(X_1, \cdots, X_n)$*, which are independently and identically distributed (IID) real d-diemsnion random variables, following a probability distribution* $p(X)$*, i.e.,* $X_i \sim p(X), X \in \mathbb{R}^d$*. We have an additional random variable* $Y$ *that is random uniform sample of these variables, i.e.,* $P(Y = X_i) = \frac{1}{n}$*. The cumulative distribution function (CDF)* $\bar{F}(t)$ *of random variable* $Y$ *will converge to the* $F(X)$*, i.e., CDF of* $X$*.*

Proof: We first define an empirical cumulative distribution function $\hat{F}_n(X)$ over the random variables $(X_1, \cdots, X_n)$:

$$\hat{F}_n(t) = \frac{1}{n} \sum_{i=1}^{n} \mathbf{1}_{X_i \leq t}, \tag{4}$$

where $\mathbf{1}_{X_i \leq t}$ is an indicator for $X_i^d \leq t^d$ for all dimensions $\{1, \cdots, d\}$.

The Glivenko–Cantelli theorem states that this empirical distribution function $\hat{F}_n(X)$ will converge to the cumulative distribution $F(X)$ if $n$ is sufficiently large:

$$\mathbf{sup}_{t \in \mathbb{R}^d} |\hat{F}_n(t) - F(t)| \to 0. \tag{5}$$

If we have an additional random variable $Y$ that its value is a random subsample of the variables $(X_1, \cdots, X_n)$:

$$P(Y = X_i) = \frac{1}{n}, \forall i = 1, 2, \cdots, n. \tag{6}$$

The CDF of this variable $\bar{F}(t)$ is:

$$\bar{F}(t) = P(Y \leq t) = \sum_{i=1}^{n} P(Y = X_i) \cdot \mathbf{1}_{X_i \leq t} = \frac{1}{n} \sum_{i=1}^{n} \mathbf{1}_{X_i \leq t} = \hat{F}_n(t). \tag{7}$$

Therefore, the CDF of $Y$ also converges to the original underlying CDF $F(t)$ if $n$ is sufficiently large.

**Proposition 2.** *Assume we have* $n$ *random samples* $(X_1, \cdots, X_n) \sim p_1$*, and another* $n$ *random samples* $(Y_1, \cdots, Y_n) \sim p_2$*, and we are also given an arbitrary bijective map between random variables, i.e.,* $\Pi : \{1, \cdots, n\} \leftrightarrow \{1, \cdots, n\}$*. If we construct a new random variable* $Z : (X, Y)$ *follows the following couplings:*

$$P(X = X_i, Y = Y_j) = \begin{cases} \frac{1}{n}, & \text{if } j = \Pi(i) \\ 0, & \text{else } j \neq \Pi(i); \end{cases}$$

*The CDF of the marginal* $P(X)$ *will converge the CDF of* $p_1$*, while the CDF of the marginal* $P(Y)$ *will converge to the CDF of* $p_2$*.*

Proof: Since $\Pi$ is bijective, we can compute the marginal $P(X = X_i)$ directly:

$$P(X = X_i) = \sum_{j=1}^{n} P(X = X_i, Y = Y_j)$$

$$= P(X = X_i, Y = Y_{\Pi(i)}) + \sum_{j \neq \Pi(i)} P(X = X_i, Y = Y_j) \tag{8}$$

$$= \frac{1}{n} + 0 = \frac{1}{n}$$

Similarly, we can show the marginal of P(Y) is also $\frac{1}{n}$. By leveraging Proposition 1, we show that $P(X)$ will converge the CDF of $p_1$, and the CDF of $P(Y)$ will converge to the CDF of $p_2$.

### A.1.2 PROOF OF OUR OT APPROXIMATION

We first give a definition of coupling $q(x_0, x_1)$ in our case before showing its marginal fullfils the marginal requirements. In particular, we denote $x_0 \in R^{N \times 3}$ and $x_1 \in R^{N \times 3}$ as two random variables following the distributions, $q_0(x_0)$ and $q_1(x_1)$, respectively. It is noted that $q_0 := N(0, I)$, which is the standard Gaussian for each dimension in $x_0$, and $q_1(x_1)$ is the distribution all possible point clouds, which involves the joint modeling of point cloud distribution given a shape $S$ ($q_1(x_1|S)$) and the distribution of shape ($q(S)$), $i.e.$, $q_1(x_1) = \int q_1(x_1|S)q(S)dS$.

We can notice that each row in $x_0$ is independently and identically distributed (IID), $i.e.$, $q_0(x_0) = \prod_i^N \hat{q}_0(x_0^i)$, where we denote the $i$-th row of $x_0$ as $x_0^i$ and distribution of $x_0^i$ as $\hat{q}_0(x_0^i)$, which is 3-dimensional unit Gaussian. We can also assume each point in $x_1$ is IID given a shape, $i.e.$, $q_1(x_1|S) = \prod_i^N \hat{q}_1(x_1^i|S)$, where we denote the $i$-th row of $x_1$ as $x_1^i$ and the distribution of $x_1^i$ as $\hat{q}_1(x_1^i|S)$.

In our superset OT precomputation for a given shape $S$, we pre-sample a set of random variables $(x_0^1 \cdots, x_0^j, \cdots, x_0^M) \sim \hat{q}_0$, and a set of random variables $(x_1^1, \cdots, x_1^k, \cdots, x_1^M) \sim \hat{q}_1$, and have a precomputed bijective mapping $\Pi : \{1, \cdots, M\} \leftrightarrow \{1, \cdots, M\}$. With these defined, our coupling $\hat{q}(x_0^i, x_1^i|S)$ for one row in the training pair $(x_0^i, x_1^i)$ given $S$ can be formulated as:

$$\hat{q}(x_0^i = x_0^j, x_1^i = x_1^k|S) = \begin{cases} \frac{1}{n}, & \text{if } j = \Pi(k) \\ 0, & \text{else } j \neq \Pi(k); \end{cases}$$

Since the each row in the training pairs are independently subsampled, the coupling of the training pair $(x_0, x_1)$ given a shape is defined as $q(x_0, x_1|S) = \prod_i^N \hat{q}(x_0^i, x_1^i|S)$. In the end, the coupling over all training pairs can be obtained by marginalize over all possible shapes, $i.e.$, $\int q(x_0, x_1|S)q(S)dS$.

**Theorem 1.** *Our coupling without blending converge the following marginal if the superset size $M$ is sufficiently large:*

$$\int q(\mathbf{x}_0, \mathbf{x}_1)d\mathbf{x}_1 = q_0(\mathbf{x}_0), \int q(\mathbf{x}_0, \mathbf{x}_1)d\mathbf{x}_0 = q_1(\mathbf{x}_1). \tag{9}$$

Proof: We first show the left constraint:

$$LHS = \int q(x_0, x_1)dx_1 = \int\int q(x_0, x_1|S)q(S)dSdx_1 \tag{10}$$

$$= \int q(S) \int q(x_0, x_1|S)dx_1 dS \qquad \text{change the order of integration} \tag{11}$$

$$= \int q(S) \int \prod_i^N \hat{q}(x_0^i, x_1^i|S)d(x_1^1, \cdots, x_1^N)dS \qquad \text{independent assumption of each row in training pair} \tag{12}$$

$$= \int q(S) \prod_i^N \int \hat{q}(x_0^i, x_1^i|S)dx_1^j dS \qquad \text{integrals of independent products} \tag{13}$$

$$= \int q(S) \prod_i^M \sum_k \hat{q}(x_0^i, x_1^k|S)dS \qquad \text{restricting to discrete values in supersets} \tag{14}$$

$$= \int q(S) \prod_i \hat{q}_0(x_0^i)dS \qquad \text{Proposition 2} \tag{15}$$

$$= \int q(S)q_0(x_0)dS = q_0(x_0) \qquad \text{independent assumption of each row in Gaussian noises} \tag{16}$$

$$\tag{17}$$

Similarly, we perform the same computation on the right constraint:

$$LHS = \int q(x_0, x_1)dx_0 = \int \int q(x_0, x_1|S)q(S)dSdx_0 \tag{18}$$

$$= \int q(S) \int q(x_0, x_1|S)dx_0 dS \qquad \text{change the order of integration} \tag{19}$$

$$= \int q(S) \int \prod_i^N \hat{q}(x_0^i, x_1^i|S)d(x_0^1, \cdots, x_0^N)dS \qquad \text{independent assumption of each row in training pair} \tag{20}$$

$$= \int q(S) \prod_i^N \int \hat{q}(x_0^i, x_1^i|S)dx_0^i dS \qquad \text{integrals of independent products} \tag{21}$$

$$= \int q(S) \prod_i \sum_j^M \hat{q}(x_0^j, x_1^i|S)dS \qquad \text{restricting to discrete values in supersets} \tag{22}$$

$$= \int q(S) \prod_i \hat{q}_1(x_1^i|S)dS \qquad \text{Proposition 2} \tag{23}$$

$$= \int q(S)q_1(x_1|S)dS = q_1(x_1) \qquad \text{independent assumption of each row in point cloud} \tag{24}$$

$$\tag{25}$$

### A.1.3 PROOF OF HYBRID COUPLING

In the last, we would like to show even with our hybrid coupling, the marginal still fulfills the requirements. In particular, we define a new noises $x_0'$ after perturbation:

$$x_0' = \sqrt{1-\beta}x_0 + \sqrt{\beta}\epsilon, \epsilon \sim N(\epsilon; 0, I), \tag{26}$$

where $\beta \in [0,1]$ is the blending coefficient. We denoted this as a conditional distribution $q(x_0'|x_0)$, which has a form of $N(x_0'; \sqrt{1-\beta}x_0, \beta)$. It is noted that since $\epsilon \sim N(\epsilon, 0, I)$, each row of $x_0'$ is also IID given $x_0$, i.e., $q_0(x_0'|x_0) = \prod_i^N \hat{q}_0(x_0'^i|x_0^i)$. Due to the independent properties, it is sufficient to show that:

$$\int q(x_0'^i, x_1^i|S)dx_0'^i = q_1(x_1^i|S), \int q(x_0'^i, x_1^i|S)dx_1^i = q_0(x_0^i). \tag{27}$$

For the sake of simplicity, we remove all the index $i$ and shape $S$ in the followings. We first show the left constraint:

$$\int q(x_0', x_1)dx_0' = \int \int q_0(x_0', x_0, x_1)dx_0 dx_0' \tag{28}$$

$$= \int \int q_0(x_0'|x_0)q(x_0, x_1)dx_0 dx_0' \tag{29}$$

$$= \int q(x_0, x_1) \int q_0(x_0'|x_0)dx_0' dx_0 \tag{30}$$

$$= \int q(x_0, x_1)(1)dx_0 \tag{31}$$

$$= q(x_1) \tag{32}$$

By Proposition 1, we can show $q(x_1)$ still converge to the right CDF if $M$ is sufficient large.

On the other hand, we show that:

$$\int q(x_0', x_1)dx_1 = \int \int q_0(x_0', x_0, x_1)dx_0 dx_1 \tag{33}$$

$$= \int \int q_0(x_0', x_0)dx_0 \tag{34}$$

$$= \int \int q_0(x_0'|x_0)q(x_0)dx_0 \tag{35}$$

$$= N(0, I) \tag{36}$$

where the last equality is obtained by inserting $q(x_0) = N(0, I)$ and $q_0(x_0'|x_0) = N(x_0'; \sqrt{1-\beta}x_0, \beta I)$.

## B IMPLEMENTATION DETAILS

**Training Details.** We implement our networks using PyTorch (Paszke et al., 2019) and run all experiments on a GPU cluster with four A100 GPUs. We employ the Adam optimizer (Kingma, 2014) to train our model with a learning rate of $2 \times 10^{-4}$ and an exponential decay of 0.998 every 1,000 iterations. Following LION (Zeng et al., 2022), we use an exponential moving average (EMA) of our model with a decay of 0.9999. Specifically, we train our unconditional generative model for approximately 600,000 iterations with a batch size of 256, taking about four days to complete. It is noted that we use larger batch sizes and more training iterations compared to existing work, including (Zhou et al., 2021; Wu et al., 2023), to effectively compare various training paradigms (Figures 6 & 7). This choice ensures our training procedure has higher stability and converges properly. Additionally, we want to highlight that the online subsampling procedure introduces negligible overhead in the training process (merely requiring additional indexing of a cached array).

**Network Architecture.** For the network architecture, we adopt the same structure as PVD (Zhou et al., 2021) and employ PVCNN (Liu et al., 2019) as our vector field network for the unconditional generation task. In the shape completion task, we use an additional 256-dimensional latent vector to represent the input partial point cloud, which is then injected into PVCNN. To do so, we use another PVCNN follow LION (Zeng et al., 2022) to extract the latent vector from the partial point cloud.

**Generated Shape Normalization.** To ensure a fair evaluation among different baselines in the unconditional task, we convert the inference results of various generative methods into a common coordinate domain. For all baseline methods, including PVD (Zhou et al., 2021), DiT-3D (Mo et al., 2023), LION (Zeng et al., 2022), and PSF (Wu et al., 2023), we respect the original normalization adopted in their training procedures. Since all these methods compute a global mean coordinate and global scale ratio across all training shapes, we use these two quantities to reverse the normalization on the generated shape, based on the values obtained from the training set. This procedure aligns with existing baselines, such as (Yang et al., 2019; Zeng et al., 2022).

## C APPROXIMATED OT

### C.1 IMPLEMENTATION DETAILS

In the following, we outline the procedure for the approximated Optimal Transport (OT) employed for supersets with size $M > 10,000$. We are motivated to use an approximated OT because the computational time for the exact method, *i.e.*, the Hungarian algorithm (Kuhn, 1955), is prohibitively large for a large superset size. Given supersets of size 10,000, it already takes 220 seconds. Considering the complexity of $O(M^3)$, this is unlikely to be scalable for larger set sizes, e.g., $M = 100,000$, for the thousands of training shapes in each ShapeNet category, even if we move this computation offline. Therefore, we resort to an approximation procedure that utilizes Wasserstein gradient flow to obtain our point superset $X_1$ by progressively transporting a noise $X_0$.

**OT Distances.** To enable an optimization procedure to move the points, we need to define an objective to be optimized, which is the 2-Wasserstein distance in our case. In particular, the OT distance can be defined as:

$$W(q_0, q_1) := \min_{q \in \Gamma(q_0, q_1)} \int C(x_0, x_1)q(x_0, x_1)dx_0 dx_1, \tag{37}$$

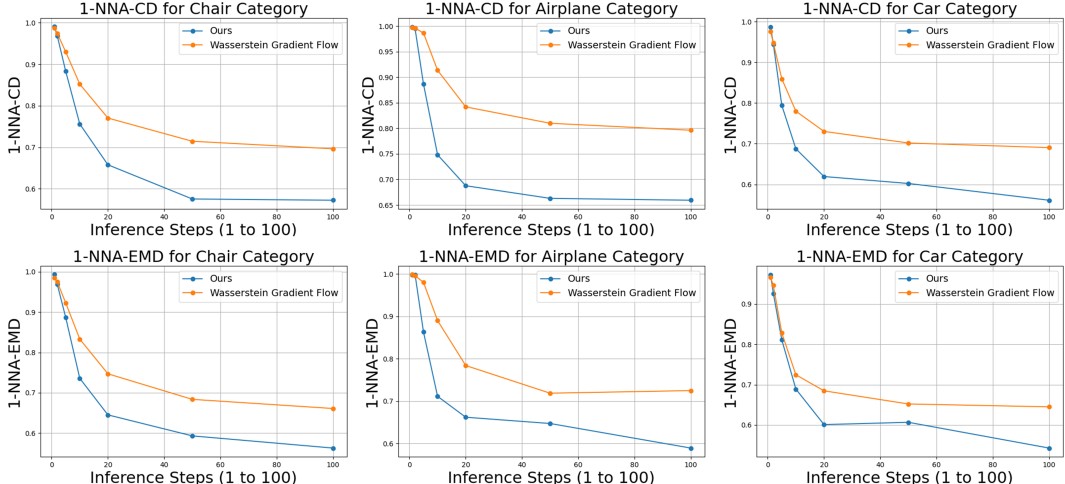

Figure 11: Quantitative comparisons of generation quality for our framework against the training pairs obtained by 1-step Wasserstein Gradient Flow. We also show 1-NNA-CD (top) and 1-NNA-EMD (bottom) for Chair (left), Airplane (middle), and Car (right). Note that a value closer to $50\%$ indicates better performance.

where $\Gamma(q_0, q_1)$ represents all couplings with marginals $q_0$ and $q_1$. However, explicitly computing the OT is not tractable, so a relaxation of this cost is introduced by adding an entropic barrier term weighted by $\epsilon$ for solving various problem, including (Léonard, 2012). This relaxation enables an efficient solution by employing the Sinkhorn algorithm, which can be highly parallelized on the GPU. As shown in (Feydy et al., 2019), this entropic barrier introduces bias in measuring the distance. Even when the marginals in the above equation are equal, i.e., $q_0 = q_1$, the distance might not be zero, leading to poor gradients. To address this issue, Genevay et al. (2018) introduce correction terms to the objective, forming the Sinkhorn divergence. Additionally, Feydy et al. (2019) provide an efficient, memory-saving GPU implementation that scales to millions of samples.

**Optimization Procedure.** With the defined objective, our optimization will compute the gradient with respect to this objective to update the current point samples. Initially, we initialize the coordinates of a set of points as the noise superset $x_0$. At each optimization iteration, we compute the Sinkhorn divergence between the current point set and the target superset $X_1'$. By repeatedly minimizing the OT distance in the form of Sinkhorn divergence, we obtain a deformed point set that fits the given superset well. Finally, we use this deformed point set as our point superset to represent the shape $S$.

Note that using this computed superset as $X_1$ introduces approximation errors compared to true samples $X_1'$. However, we empirically observe that the procedure usually converges well with only small errors, and in our experiments, the large superset size typically outweighs the introduced error (as shown in Section 4.3).

### C.2 TRAINING PAIRS BY 1-STEP APPROXIMATION

We also compare our approach with training pairs obtained through the optimization procedure described above. In this experiment, we perform only a single optimization iteration on point clouds and noises of size $2,048$ and set $\beta$ to $0.0$. For each training iteration, we use the optimized results, $X_0$ and $X_1$, as our training pair. The quantitative comparison results are shown in Figure 11. Our approach consistently outperforms the 1-step Wasserstein Gradient Flow across all categories and most inference timesteps.

## D ADDITIONAL QUANTITATIVE COMPARISON

### D.1 COVERAGE METRIC (COV)

Following Zhou et al. (2021); Zeng et al. (2022), we also evaluate coverage (COV), a metric that measures the diversity of generated 3D shapes. COV calculates the proportion of testing shapes that

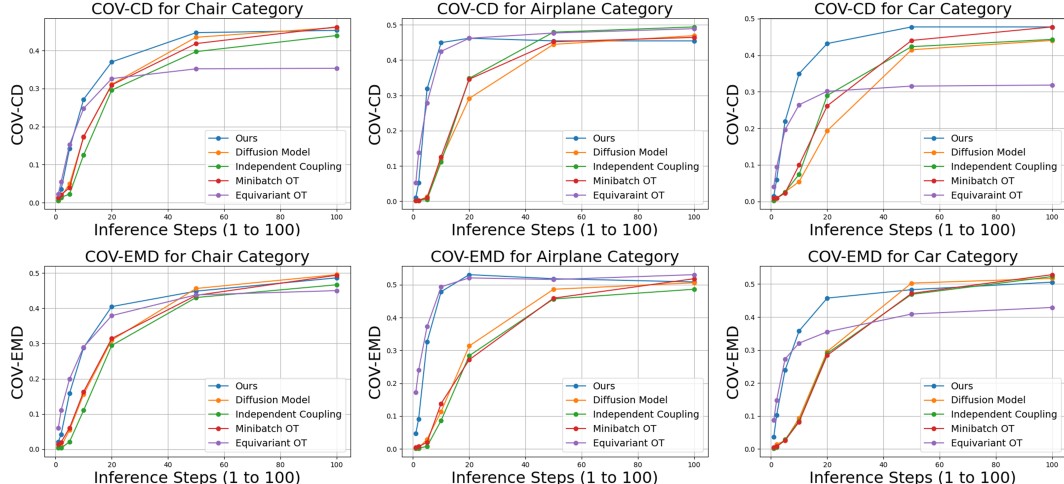

Figure 12: Quantitative comparisons of generation diversity for different training paradigms using COV-CD (top) and COV-EMD (bottom) for Chair (left), Airplane (middle), and Car (right). We present evaluation metrics across various inference steps, *i.e.*, from 1 steps to 100 steps, for five methods: (i) ours, (ii) diffusion model with v-prediction (Salimans & Ho, 2022), and three flow matching models with different coupling methods: (iii) independent coupling (Lipman et al., 2022), (iv) Minibatch OT (Tong et al., 2023; Pooladian et al., 2023), and (v) Equivariant OT (Song et al., 2024; Klein et al., 2024). Note that a higher value indicates better performance.

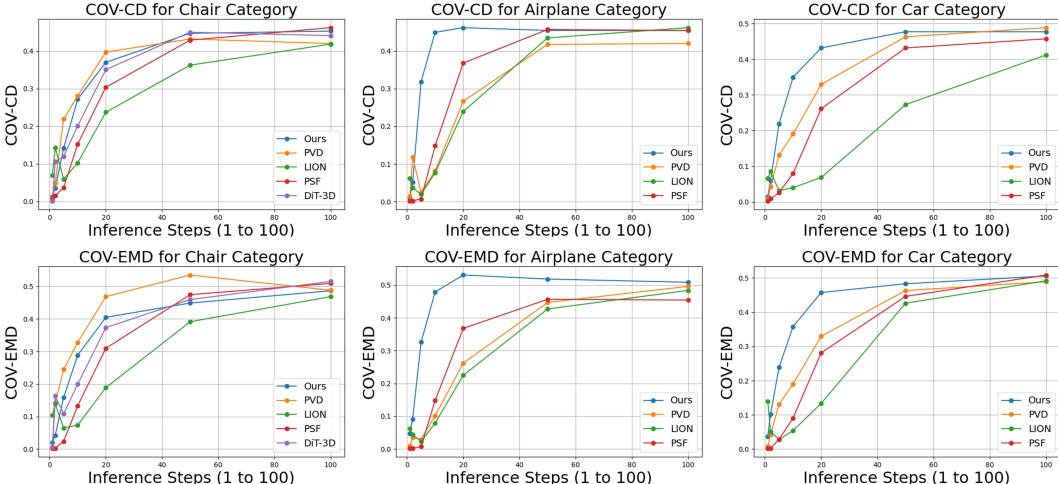

Figure 13: Quantitative comparisons with other point cloud generation methods using the COV-CD (top) and COV-EMD metrics (bottom) for Chair (left), Airplane (middle), and Car (right). We present evaluation metrics across various inference steps, *i.e.*, from 1 step to 100 steps, for five methods: (i) ours, (ii) PVD (Zhou et al., 2021), (iii) LION (Zeng et al., 2022), (iv) PSF (Wu et al., 2023) without rectified flow, and (v) DiT-3D (Mo et al., 2023).

can be retrieved by generated shapes, with higher values indicating higher diversity. However, Yang et al. (2019); Zhou et al. (2021); Zeng et al. (2022); Wu et al. (2023) have noted that this metric is not robust as training set shapes can have worse COV than generated results. Moreover, Yang et al. (2019) suggest that perfect coverage scores are possible even when distances between generated and testing point clouds are arbitrarily large. Given these limitations, COV should be considered only as a reference metric, while 1-NNA provides a more reliable measure that captures both generation quality and diversity.

**Evaluation Results.** We present quantitative comparisons with different baselines in Figure 12 and 13. Our approach generates shapes with reasonable diversity even with a limited number of steps (10-20), demonstrating the framework's effectiveness. With sufficient inference steps (100), our approach achieves comparable performance to other baselines. However, we observe contradictory conclusions between 1-NNA and COV metrics (as shown by simultaneously high 1-NNA and COV

Table 3: We provide evaluation results (1-NNA-CD and 1-NNA-EMD) using 1000 inference steps for DiT-3D (Mo et al., 2023), PSF (Wu et al., 2023) without rectified flow, PVD (Zhou et al., 2021), and LION (Zeng et al., 2022). The best-performing method is highlighted in red, while the second-best is shown in blue.

|  | Chair | | Airplane | | Car | |
|---|---|---|---|---|---|---|
|  | 1-NNA-CD | 1-NNA-EMD | 1-NNA-CD | 1-NNA-EMD | 1-NNA-CD | 1-NNA-EMD |
| DiT-3D | 0.6072 | 0.5604 | - | - | - | - |
| PSF | 0.5612 | 0.5642 | 0.7457 | 0.6617 | 0.5682 | 0.5412 |
| PVD | 0.5626 | 0.5332 | 0.7382 | 0.6481 | 0.5455 | 0.5383 |
| LION | 0.537 | 0.5234 | 0.6741 | 0.6123 | 0.5341 | 0.5114 |
| Ours | 0.5551 | 0.5763 | 0.6864 | 0.6185 | 0.5966 | 0.5355 |

scores for Equivariant OT in the Airplane category), which aligns with the previously discussed limitations of the COV metric.

## D.2 MORE INFERENCE STEPS

In addition to the baseline comparisons presented in the main paper (Figures 8 and 9), we provide additional comparisons with 1000 inference steps, matching the original settings used by the baseline methods. We employ the DDPM sampler (Ho et al., 2020) rather than the DDIM sampler (Song & Ermon, 2019) in this experiment and refer to the original values of PVD and LION reported in (Zeng et al., 2022).

**Evaluation Results.** We present the evaluation results based on 1-NNA-CD and 1-NNA-EMD in Table 3. Our method achieves comparable performance with PVD (Zhou et al., 2021), which also directly generates point clouds. When compared to LION (Zeng et al., 2022), which generates latent point cloud representations, our framework performs slightly worse. Note it is known that at a high sampling budget, SDE-based samplers often outperform ODE-based samples (see (Karras et al., 2022) and (Xu et al., 2023)).

## E ADDITIONAL VISUAL RESULTS

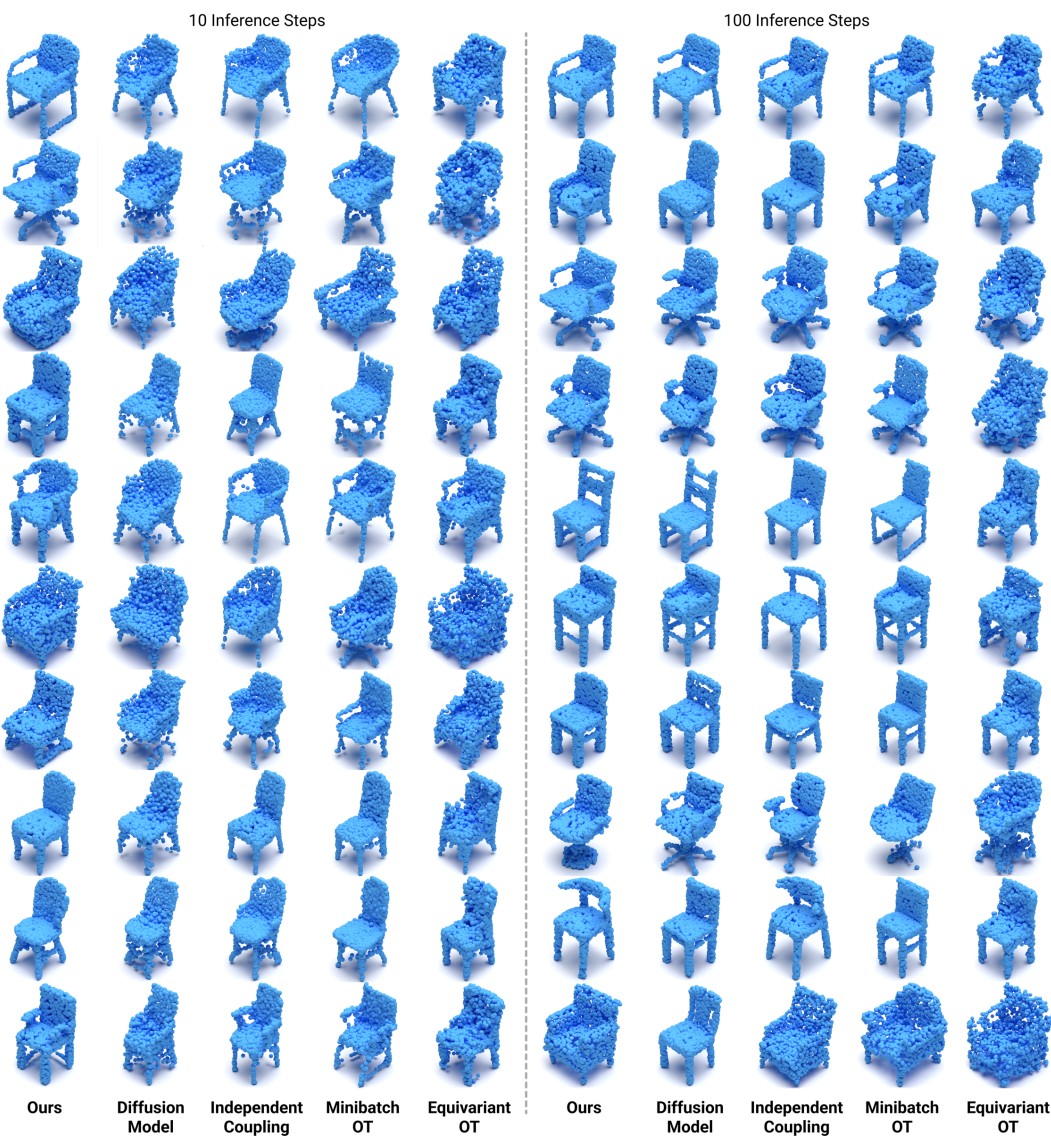

Figure 14: Qualitative comparisons of generation quality for Chair category. We present inference results with 10 steps (left) and 100 steps (right).

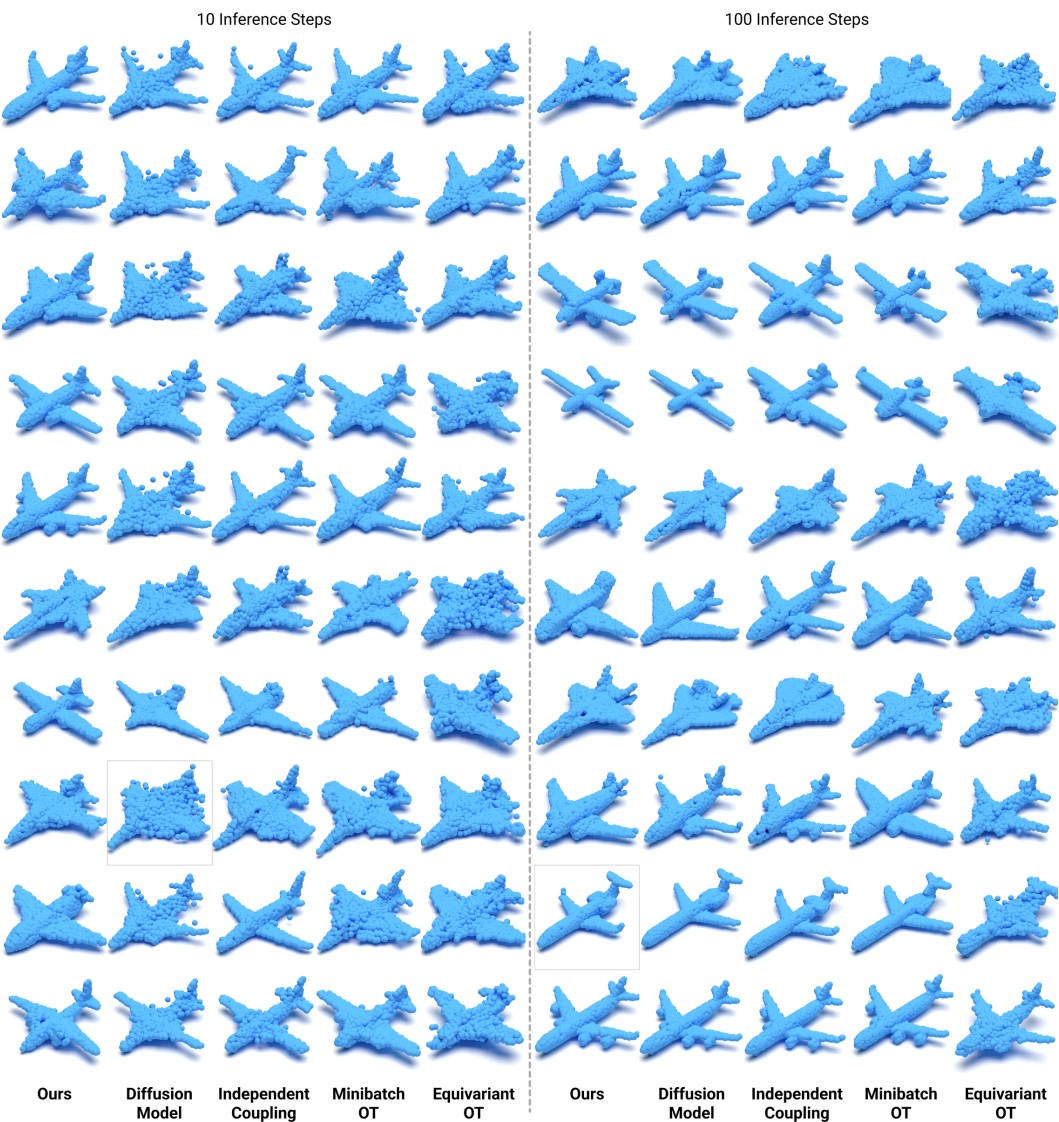

Figure 15: Qualitative comparisons of generation quality for Airplane category. We present inference results with 10 steps (left) and 100 steps (right).

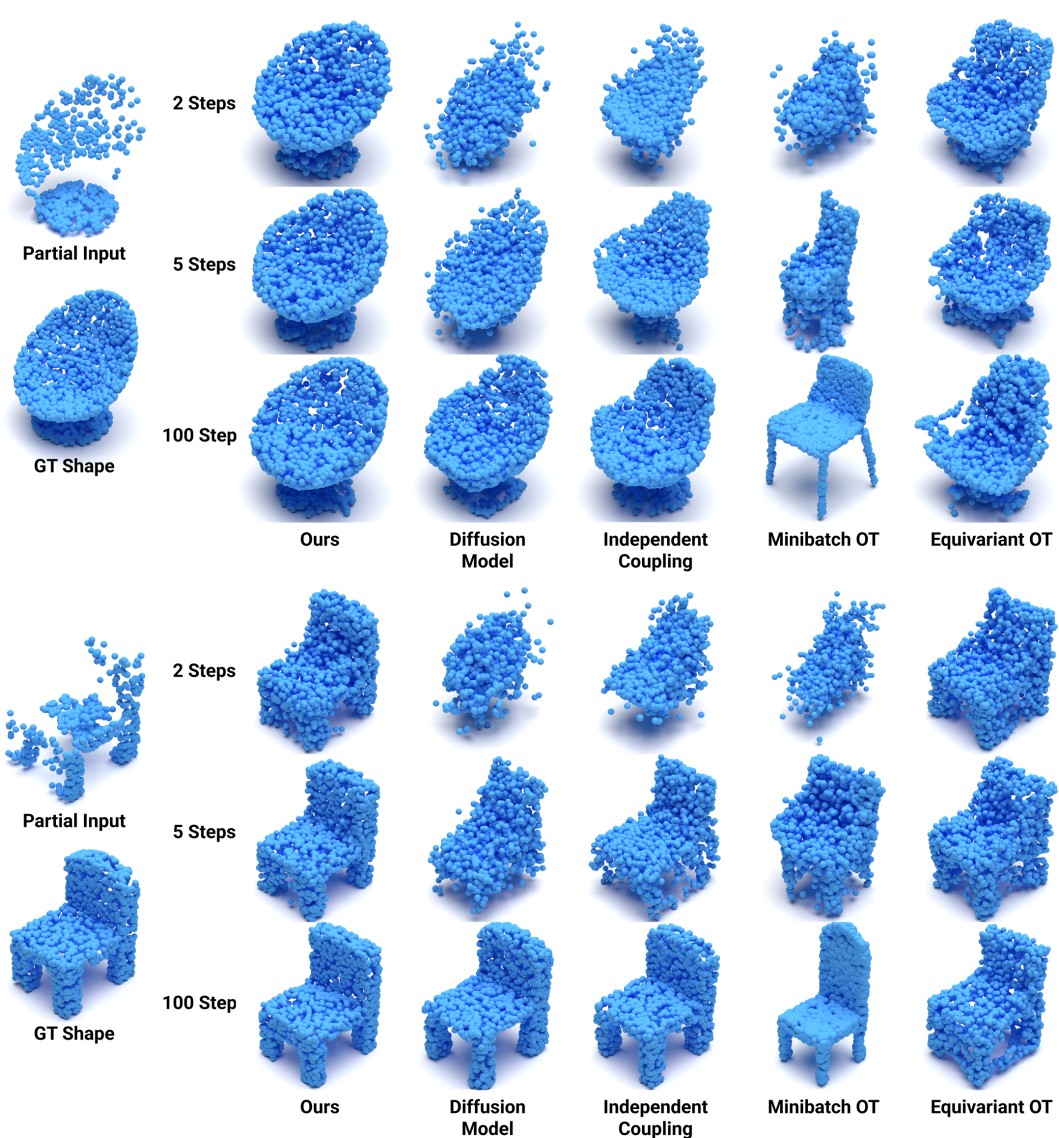

Figure 16: Qualitative comparisons with other methods show the completion generated by 2, 5, and 100 steps, respectively.

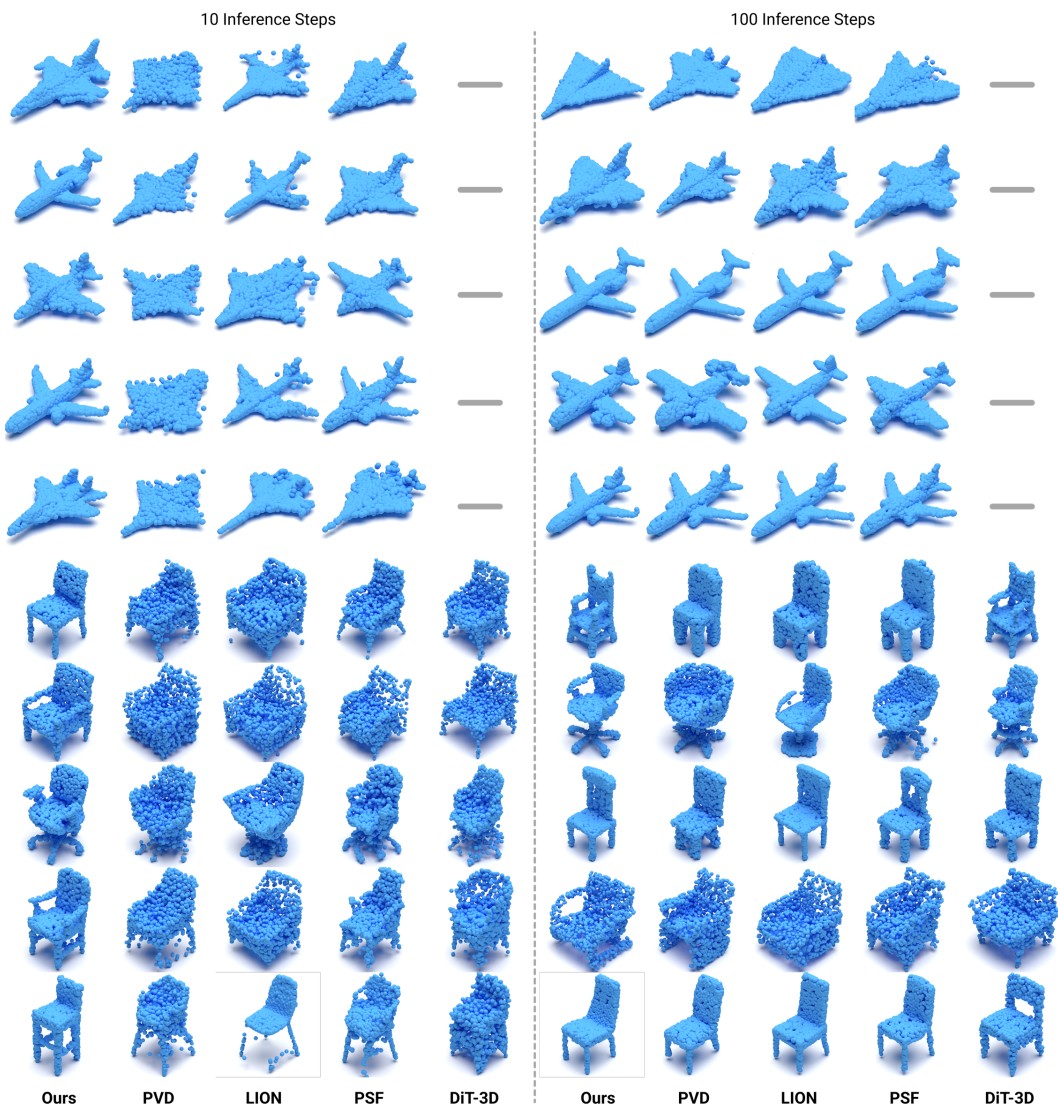

Figure 17: Qualitative comparisons of generation quality for Airplane (top) and Chair (bottom) categories. We present inference results with 10 steps (left) and 100 steps (right).

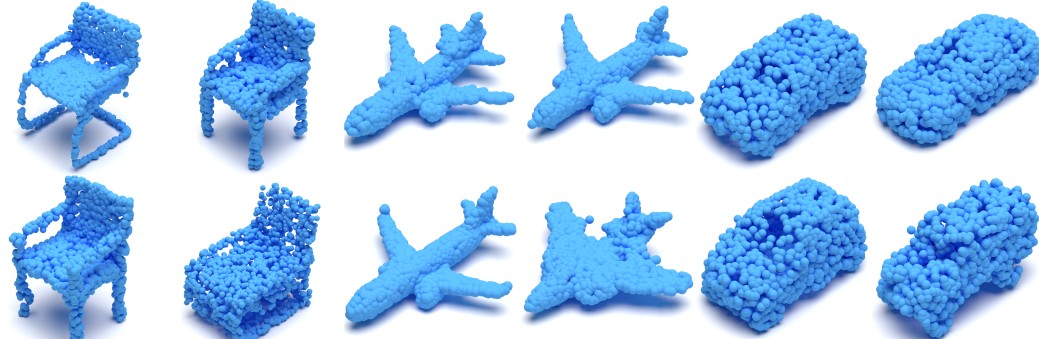

Figure 18: We provide the generation results from a model trained with shapes from all three categories (Chair, Airplane and Car) together. The results indicate that our approach can achieve high-quality generation with limited inference steps (20 steps) even when training with a mixture of shapes from different categories.

