# OpenReview forum: "Not-So-Optimal Transport Flows for 3D Point Cloud Generation"
_ICLR.cc/2025/Conference — ICLR 2025 Poster_

### Official Review · Reviewer_o98M · 2024-11-02

**Soundness:** 3
**Presentation:** 3
**Contribution:** 3
**Rating:** 6
**Confidence:** 3

**Summary:**

This paper presents a novel flow matching methodology for 3D point cloud generation. To overcome the limitations of existing Optimal Transport (OT) based methods (scalability issues, complex flow learning), it introduces 'not-so-optimal' transport flow matching. The proposed method enables efficient learning by combining offline superset OT computation with online subsampling, and reduces flow model complexity through a hybrid approach with independent coupling. The paper makes several significant contributions to the field of 3D point cloud generation. First, it provides a thorough analysis of existing OT-based methods, meticulously identifying their limitations in terms of scalability and computational overhead. Building on this analysis, the authors introduce an innovative approach combining superset OT precomputation with efficient online subsampling, addressing the identified scalability issues. They further enhance their methodology by proposing a hybrid coupling approach that cleverly combines OT and independent coupling, offering a more balanced solution. The effectiveness of these contributions is demonstrated through state-of-the-art performance on the ShapeNet benchmark, showing superior results in both unconditional generation and shape completion tasks.

**Strengths:**

1. The paper demonstrates balance between theoretical foundations and empirical validation. The authors provide mathematical analysis of their approach while supporting their claims with comprehensive experimental results across multiple benchmarks and metrics.

2. The authors systematically analyze existing methods' limitations, particularly in terms of scalability and computational complexity. They not only identify these challenges clearly but also propose concrete, well-thought-out solutions that directly address each limitation.

3. The authors show remarkable approach in managing the inherent trade-off between computational resources and model performance. Their proposed hybrid approach effectively balances the benefits of optimal transport with the computational efficiency of independent coupling, resulting in a practical solution.

**Weaknesses:**

1. While the paper addresses permutation invariance in detail, it lacks comprehensive treatment of other important invariances in 3D point cloud generation, particularly rotational invariance.

2. The paper provides insufficient theoretical guidance for determining optimal superset size M and hybrid coupling's $\beta$ parameter. While empirical results are presented for various superset sizes, blending coefficients. Without robust theoretical foundations for these choices, it becomes challenging to establish meaningful connections with existing theoretical frameworks and related research domains.

3. The experimental validation focuses primarily on 3D ShapeNet datasets and benchmarks, with limited exploration of more challenging real-world applications. The lack of validation on complex domains like large molecular structures or protein configurations leaves questions about the method's broader applicability and scalability in these important areas.

**Questions:**

1. Is there a theoretical foundation for determining the optimal superset size in superset OT precomputation?

2. How should the value of $\beta$ in hybrid coupling vary depending on dataset characteristics and task requirements?

3. Can the proposed method be effectively applied to other types of point cloud data?

---

### Official Review · Reviewer_mmfz · 2024-11-03

**Soundness:** 3
**Presentation:** 3
**Contribution:** 3
**Rating:** 6
**Confidence:** 3

**Summary:**

This paper explores optimal transport (OT) flow for point cloud generation, finding that existing OT approximations are not directly applicable to this task. The authors suggest that this limitation arises because equivariant OT flows must learn a complex, high-Lipschitz function early in the generation process. To address this, they introduce a "not-so-optimal" transport flow that combines offline superset OT precomputation with online subsampling and propose a hybrid coupling strategy.

**Strengths:**

1. The paper is well-written and easy to follow.
2. Extensive experiments are conducted to support the claims in the paper.
3. Experiment results demonstrate the effectiveness of the proposed methods, especially with fewer inference steps.

**Weaknesses:**

1. 1-NNA CD and EMD are mainly used to measure the quaility. However, another aspect of generation, the diversity, has been ignored in the experiment. Coverage (COV) with CD and EMD should be reported to measure the generation diversity. You can refer to DiT-3D for details of COV.
2. The experiments are focused solely on single-category generation. It would be more valuable to test the method on multi-category training, such as using the full ShapeNet-13, ShapeNet-55 or Objaverse dataset.

**Questions:**

1. How is the performance of the proposed method if the time steps reach 1000 as in other baselines? Will it also achieve better results against the baselines?
2. In Figure 4, “Note that we subsample the point cloud to 30 points for a better trajectory visualization”, how many points are used for training in these visualization experiments, 30 points or more?
3. In Line 374, what does ‘hyperparameters’ refer to?

---

### Official Review · Reviewer_GZj5 · 2024-11-04

**Soundness:** 4
**Presentation:** 3
**Contribution:** 4
**Rating:** 8
**Confidence:** 3

**Summary:**

The paper designs an optimal transport flow matching method for 3D point cloud generation, addressing the critical challenge of permutation invariance in point clouds. The method incorporates offline OT mapping between data and noise to reduce training time. Additionally, it employs a hybrid coupling strategy that blends independent coupling with optimal transport to improve the alignment of point clouds. The authors provide a theoretical explanation and empirical evidence demonstrating why traditional OT methods struggle with point clouds, showing that their proposed approach effectively overcomes these limitations.

**Strengths:**

- Novelty: The adaptation of optimal transport methods in the context of point cloud generation is a significant and novel contribution. This approach addresses the permutation invariance of point clouds in flow-matching-based point cloud generation.

**Weaknesses:**

- Complex Computation and Slow Training Speed: Despite the use of offline OT matching, the training process remains computationally intensive due to the random subsampling of data-noise pairs and the iterative training of the vector field​. This results in significant training time, with approximately four days required on a cluster with four A100 GPUs, highlighting the method's complex computation and slow speed issues.

- Scalability Issues: The use of Wasserstein gradient flow and the Hungarian algorithm for optimal transport computation in large-scale point clouds is computationally expensive. Additionally, the method necessitates separate training for each category, which not only diminishes efficiency and scalability but also requires extensive training time for each individual category. This lengthy training process further exacerbates the overall computational burden. Compared to contemporary 3D generation approaches that can efficiently handle multi-category generation, this method does not scale well.

**Questions:**

- Could you please clarify how Figure 1 effectively illustrates the different coupling types between Gaussian noise and point clouds?
- How does your approach handle noisy input data, particularly in scenarios where the data may contain outliers?
- What strategies do you plan to implement to address scalability issues in practical applications?

---

### Official Review · Reviewer_F3TG · 2024-11-04

**Soundness:** 4
**Presentation:** 4
**Contribution:** 4
**Rating:** 8
**Confidence:** 4

**Summary:**

The paper proposes a paradigm for training a flow-based generative model for permutation-invariant data such as 3D point clouds using a simple and efficient approximation of optimal transport (OT).
Computing the optimal transport flows online scales poorly to a large number of points due to the prohibitive cost, while existing works based on approximations perform poorly.
On the contrary, the proposed method precomputes Gaussian-to-points OT of point clouds offline, and subsample it online to form the training pairs.
Apart from the OT approximation scheme, the paper also uncovers the issue regarding high Lipchitz at $t=0$, and proposes adding small Gaussian noise during training as a remedy.
The proposed method is benchmarked on ShapeNet for point cloud completion and unconditional generation, outperforming existing diffusion and flow-based approaches.

**Strengths:**

* The proposed method achieves top performance among approximate OT flow and diffusion baselines, especially in the low-iteration regime.
* The paper is well written, and the analysis on the behavior of the proposed approximation is very comprehensive.
* It is surprising yet convincing to see that a more optimal OT leads to poorer performance due to high Lipchitz.

**Weaknesses:**

* The proposed method still requires the computation of a dense OT offline. The computational cost can still be very high for large point clouds. I wonder what is the number of points used for precomputing the OT superset, and how long does it take to process one shape?

**Questions:**

* I wonder if it is possible to use an even worse (but fast) OT approximation algorithm (such as Feydy et al., 2019 with fewer iterations) to replace the hybrid coupling and enable efficient online sampling? Would it achieve the same purpose as the proposed Gaussian noise perturbation?
* How critical is the size of the precomputed superset in terms of the model performance?

---

### Official Review · Reviewer_gCWN · 2024-11-07

**Soundness:** 3
**Presentation:** 3
**Contribution:** 2
**Rating:** 6
**Confidence:** 4

**Summary:**

This paper introduces an offline superset OT pre-computation method followed by an efficient online subsampling to reduce the complexity of target flow models which is hard to be approximated by the neural networks. The proposed framework could achieve good shape generation with a few steps.

**Strengths:**

This paper could generate fine 3D point results within limited steps.

**Weaknesses:**

1. Energy-based models, such as [1][2], are naturally permutation-invariant with respect to the order of point cloud data. However, these models lack sufficient discussion and comparative analysis, which would provide a clearer understanding of their strengths and limitations with the proposed method.
2. The author asserts that diffusion models lack permutation-invariance in point cloud generation. However, recent studies, including [3], which use point-voxel representations; [4], which incorporate translation- and rotation-invariant features; and [5], which leverage latent diffusion models, are not included in the baselines for comparison.
3. The author claims that the proposed method achieves high-quality generation with a limited number of inference steps. However, other fast sampling methods, such as [6], are not considered, which would offer a broader perspective on the efficiency of sampling approaches.
4. While the author suggests that the proposed method scales well, there is no study on its performance across varying resolutions of 3D shapes. Furthermore, high-resolution 3D point generation methods, such as [7] and [8], are not included, which limits the scope of comparison for resolution-dependent generation quality.

[1] Xie, Jianwen, et al. "Generative pointnet: Deep energy-based learning on unordered point sets for 3d generation, reconstruction and classification." Proceedings of the IEEE/CVF Conference on Computer Vision and Pattern Recognition. 2021.

[2] Xie, Jianwen, et al. "Generative VoxelNet: Learning energy-based models for 3D shape synthesis and analysis." IEEE Transactions on Pattern Analysis and Machine Intelligence 44.5 (2020): 2468-2484.

[3] Zhou, Linqi, Yilun Du, and Jiajun Wu. "3d shape generation and completion through point-voxel diffusion." Proceedings of the IEEE/CVF international conference on computer vision. 2021.

[4] Peng, Yong, et al. "SE (3)-Diffusion: An Equivariant Diffusion Model for 3D Point Cloud Generation." International Conference on Genetic and Evolutionary Computing. Singapore: Springer Nature Singapore, 2023.

[5] Zhao, Runfeng, Junzhong Ji, and Minglong Lei. "Decomposed Latent Diffusion Model for 3D Point Cloud Generation." Chinese Conference on Pattern Recognition and Computer Vision (PRCV). Singapore: Springer Nature Singapore, 2024.

[6] Wu, Lemeng, et al. "Fast point cloud generation with straight flows." Proceedings of the IEEE/CVF conference on computer vision and pattern recognition. 2023.

[7] Huang, Zixuan, et al. "PointInfinity: Resolution-Invariant Point Diffusion Models." Proceedings of the IEEE/CVF Conference on Computer Vision and Pattern Recognition. 2024.

[8] Wen, Xin, et al. "Point cloud completion by skip-attention network with hierarchical folding." Proceedings of the IEEE/CVF conference on computer vision and pattern recognition. 2020.

**Questions:**

1. Could the author provide a broader range of inference steps in Figure 8? Additionally, is there a comparison available for the models when they have converged?
2. Why is rotational invariance not considered or discussed in the paper?

---

### Public Comment · ~Bi'an_Du1 · 2024-11-29
**Reproduction of this work**

Nice work! Will you release your codes & checkpoints upon acceptance of the paper?

---

### Meta-Review · Area_Chair_9sBc · 2024-12-19

**Metareview:**

This paper explores point cloud generation by proposing not-so-optimal transport flow models that obtain an approximate OT by an offline OT precomputation, enabling an efficient construction of OT pairs for training. Extensive empirical studies demonstrate that the proposed model outperforms existing diffusion-based and flow-based methods across a wide range of tasks, including unconditional generation and shape completion on the ShapeNet benchmark. The paper is well-organized, well-written, and presents appealing results with a novel method. The revision effectively addresses the concerns raised by the reviewers. However, one notable weakness is the lack of comparison with energy-based point cloud generation models. The AC recognizes the novelty of the proposed framework and its promising results. After the rebuttal, all reviewers leaned toward accepting the paper. The AC concurs with the reviewers and recommends the paper for acceptance. To further enhance the quality of the paper, the AC encourages the authors to incorporate the reviewers' suggestions in the final revision.

**Additional Comments On Reviewer Discussion:**

All reviewers have reached a consensus to accept the paper, as the rebuttal effectively addressed the major concerns they raised. They agree that the paper is novel and that the experiments sufficiently demonstrate the effectiveness of the proposed model.

---

### Decision · Program_Chairs · 2025-01-22

Accept (Poster)